# Biomass Production and Removal of Nitrogen and Phosphorus from Processed Municipal Wastewater by *Salix schwerinii*: A Field Trial

**Muhammad Mohsin** [1,*] [ID]**, Erik Kaipiainen** [1]**, Mir Md Abdus Salam** [1,2]**, Nikolai Evstishenkov** [1]**, Nicole Nawrot** [3] [ID]**, Aki Villa** [1]**, Ewa Wojciechowska** [3] [ID]**, Suvi Kuittinen** [1] **and Ari Pappinen** [1,*]

1   School of Forest Sciences, University of Eastern Finland, Yliopistokatu 7, P.O. Box 111, 80100 Joensuu, Finland; erik.kaipiainen@uef.fi (E.K.); mir.salam@uef.fi (M.M.A.S.); nikolai.evtishenkov@uef.fi (N.E.); aki.villa@uef.fi (A.V.); suvi.kuittinen@uef.fi (S.K.)
2   Natural Resources Institute Finland (LUKE), Yliopistokatu 6, 80100 Joensuu, Finland
3   Faculty of Civil and Environmental Engineering, Gdansk University of Technology, 80-233 Gdansk, Poland; nicole.nawrot@pg.edu.pl (N.N.); ewa.wojciechowska@pg.edu.pl (E.W.)
*   Correspondence: muham@uef.fi (M.M.); ari.pappinen@uef.fi (A.P.)

**Abstract:** In many Baltic regions, short-rotation willow (*Salix* spp.) is used as a vegetation filter for wastewater treatment and recycling of valuable nutrients to upsurge bioeconomy development. In this context, a four-year field trial (2016–2019) was carried out near a wastewater treatment plant in eastern Finland (Outokumpu) to investigate the effect of the processed wastewater (WW) on biomass production as well as the nutrients uptake capability (mainly N and P) by a willow variety (*Salix schwerinii*). Results indicated that WW irrigation expressively increased the willow diameter growth and biomass yield around 256% and 6510%, respectively, compared to the control treatment site (without WW). The willow was also able to accumulate approximately 41–60% of the N and 32–50% of the P in two years (2018–2019). Overall, willow showed a total 20% mortality rate under WW irrigation throughout the growing periods (2017–2019) as compared to control (39%). The results demonstrate that willow has the potential to control eutrophication (reducing nutrients load) from the wastewater with the best survival rate and can provide high biomass production for bioenergy generations in cold climatic conditions.

**Keywords:** water pollution; wastewater reuse; nutrient; nature-based solution; willow

## 1. Introduction

Nitrogen (N) and phosphorus (P) are essential nutrients for all living organisms, but higher concentrations can exert a negative impact on the ecosystem. Technologies to recover essential nutrients from the processed wastewater have become highly essential due to the increase in fertilizers prices and strict discharge limits on excessive mineral nutrients [1]. The overall sustainability of wastewater treatment plants can be upgraded by reducing the use of non-renewable resources, minimizing waste generation, and implementing resource recycling approaches [2]. In many countries, municipal wastewater contains a sufficient amount of P and N and is recognized as a valuable resource. It could be favorable for both agroforestry sectors and achieving food security once it has been properly treated. Therefore, wastewater can be a valuable source of organic fertilizer if effective recovery processes are adopted [3].

In Finland, all the municipal wastewater facilities are required to remove P and N as per the legal limits. According to Finnish Environmental Institute [4], the Finnish rivers carry an annual average of 74,000 tons of N and 3400 tons of P into the Baltic Sea, which mainly comes from agricultural effluents and domestic wastewater. Long-term sustainability and nutrient self-sufficiency can be achieved if these nutrients in the wastewater are recycled using a nature-based and cost-friendly method such as phytoremediation [5].

Phytoremediation is an eco-friendly and cost-effective method for remediating polluted soils and wastewater by fast-growing and high biomass producing plants to accumulate metals/nutrients from the soil and water into plant biomass [6]. It has been reported that the resulting biomass after remediation can be used for energy and heat generations in biorefineries and could become an environmentally friendly form of biotechnology and promote bioeconomy development [7]. Further, plant-associated microorganisms commonly named "phytomicrobiome", for example, epiphytes, endophytes, root microbiome, and phyllosphere microbiota, as well as bacteria and fungi, can be used in agriculture, pharma, and medicinal industries for environmental protection. These microbiomes produce divergent bioactive molecules that support the metabolization of greenhouse gases and metals and hydrocarbons degradation [8].

Previously, recovery of up to 650 and 100 kg/ha of N and P, respectively, has been reported for annual and woody species under processed municipal wastewater (WW) irrigations [9]. Salam et al. [10,11] have also reported a significant accumulation of N and P by two willow cultivars, including *Salix schwerinii* and Klara (*Salix viminalis x Salix schwerinii x Salix dasyclados*) irrigated with WW under greenhouse conditions.

Willows (*Salix* spp.) is considered an effective tool to eliminate the heavy metals/metalloids from polluted sites [12,13]. In cold climatic regions such as Finland, the use of short-rotation willow to treat WW presents a potential solution and an innovative way of wastewater and polluted groundwater purification as well as nutrient recovery (N and P) that require low construction and operating costs. Willows have unique characteristics such as fast growth in boreal climate, high biomass production, bioenergy potential, remarkable metabolic and absorption capabilities, deep-rooted system, high evapotranspiration rate, and ability to uptake a variety of pollutants and nutrients into their biomass, which make them suitable for numerous environmental applications [14,15].

Furthermore, phytoremediation could also be used to remediate eutrophicated water (high level of N and P), which has recently become global water pollution, including Finland, due to the rapid industrialization, urbanization, and excessive agricultural practices. Eutrophicated water has ecological consequences on aquatic ecosystem functions, processes, and structures such as fast growth of algae and other phytoplankton deteriorate the water quality and subsequently affect the sustainable use of water resources [16]. The Gulf of Finland (Baltic Sea) and many Finnish lakes are suffering from eutrophication, which is considered an emerging domestic problem. In addition, current Finland's N and P levels have been doubled from the previous levels. To tackle this problem, nutrient runoff must be reduced more efficiently, particularly in the agriculture and industrial sectors [17].

The use of WW in agriculture could provide an alternative source of freshwater use for irrigation and could also be an additional source of nutrients and organic matter. It can also modify the minerals, macro-and micronutrients for plant growth, soil pH, soil buffer capacity, and cation exchange capacity [18]. Globally, processed municipal or industrial wastewater is used for the irrigation of about >20 million ha of crops worldwide. In Mexico, around 70,000 ha of agricultural lands are irrigated with WW, and more than 75% of the WW is used for crop irrigation in the USA [19]. Furthermore, the reuse of wastewater for crop irrigation could contribute to mitigating water shortage, support the agriculture sector and protect groundwater resources [20], and can also enhance the economic benefits for farmers due to the reduced need for fertilizer [21].

To the best of our knowledge, many studies have been focused on untreated wastewater, but very little attention has been paid concerning the effects of WW irrigation on short-rotation crops such as willow as the results would provide insight about re-use of wastewater for crops irrigation to reduce pressure on freshwater usage and dependency on chemical fertilizers. We hypothesize that willow plantations would accumulate the highest amount of nutrients from the WW and produce high biomass yield under WW irrigation. Therefore, this study aims to examine the effects of WW on the growth diameter of willow and total dry biomass production as well as total N and P accumulation by willow once exposed to WW irrigation.

## 2. Materials and Methods

### 2.1. Study Site and Experimental Setup

A field trial was established adjacent to a municipal wastewater treatment plant in Jokipohja Outokumpu (62°42′44.6″ N 29°02′58.5″ E) Eastern Finland. The Jokipohja plant is characterized by initial mechanical treatment of wastewaters, chemical removal of P with ferrous sulfate ($FeSO_4$), polymers and lime (CaO), and secondary treatment of activated sludge. The local stainless steel metal industry and the Jyri landfill area of the Outokumpu also transfer their wastewaters to the plant for purification. The wastewater plant has a capacity for an organic load from approximately 10,000 inhabitants, and the average flow was observed around 3750 $m^3$ $day^{-1}$. The biological oxygen demand (BOD) and chemical oxygen demand (COD) values of untreated wastewater varied from 490 to 630 mg $L^{-1}$ and 1200 to 1500 mg $L^{-1}$, respectively.

The experimental plots in Outokumpu were set up in a 3000 $m^2$ area close to the Lahdenjoki river. Another experimental plot in Siikasalmi (62°30′43.8″ N 29°21′44.2″ E) with an area of 3000 $m^2$ was considered as a reference control. Both experimental plots are shown in Figure 1. The soil properties from both study areas were identical. Cuttings of willow (length: 20 cm; diameter: 15 mm; weight: 10 g) were collected from the Siikasalmi experimental area of the University of Eastern Finland. Willow cuttings were planted manually to the soil in June 2016 in pair rows where the row spacing was 1 m, and the spacing between the pair rows was 150 cm. Rows were covered with black plastic to prevent weeds from growing. The planting density of the willow was around 24,000 per hectare.

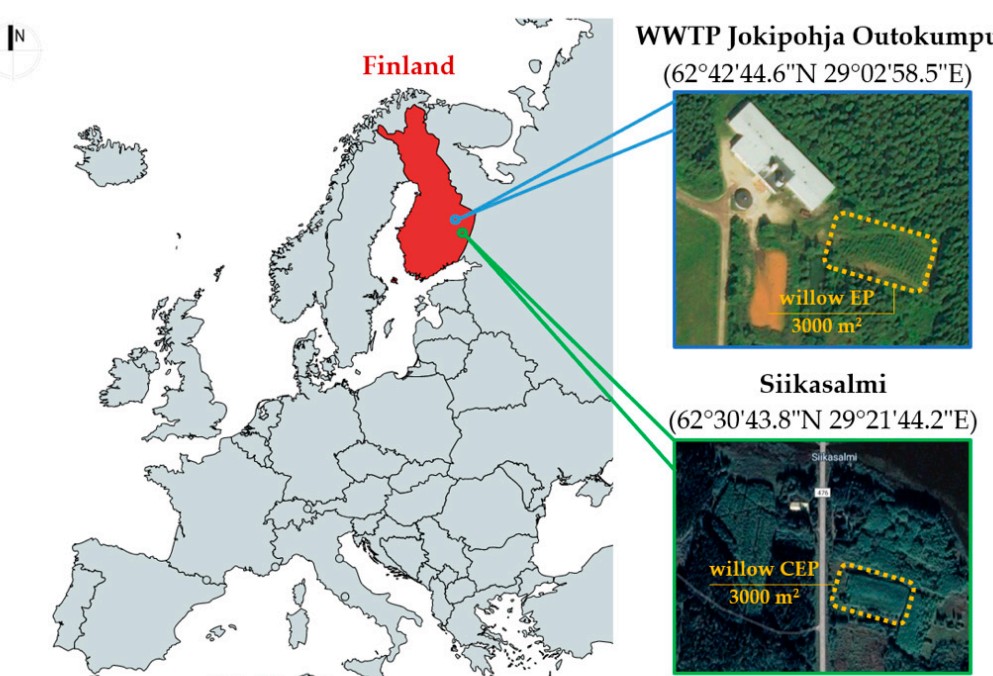

**Explanations:** EP – experimental plot; CEP – control experimental plot

**Figure 1.** The location of wastewater treatment plant (WWTP) and willow experiment plot (EP) in Jokipohja Outokumpu and control experimental plot (CEP) in Siikasalmi.

### 2.2. Scheme of Processed Wastewater (WW) Irrigation and Weather Conditions

The flow of processed wastewater (WW) from the wastewater plant was carried out by automatic systems. WW irrigation to the field was delivered by a specially adjusted pipe connected to the mainstream pipe (Figure 2). The water flow meter was installed to monitor the water load to the experimental field. The pipes were installed after the surface was leveled out, and the WW from the wastewater plant was pumped to the experimental

field, where the WW was then distributed with thinner pipe outlets. The overflow water that was not absorbed by the field drifted on the tilted surface of the field toward Lahenjoki to a collector ditch made by an excavator, and this ditch was converted into a basin. A V-dam was constructed in the lower corner of that basin. The amount of pumped WW was noted from a water strider installed to the wastewater treatment plant. The amount of the WW outflow from the field was measured from the V-dam, and at the same time, the water samples were taken. WW runoff around the field was not able to reach the field but was gathered to collector ditches. These ditches drained from the eastside of the field to Lahenjoki.

**Figure 2.** The scheme of irrigation system in Jokipohja Outokumpu experiment plot with willow.

### 2.3. Processed Wastewater Flow and Outflow to the Field and Nutrients Load

WW irrigation to willow plantations was provided at a rate of 10 mm day$^{-1}$ based on the Swedish model [14]. The cumulative reference evapotranspiration (mm) from May to September was estimated at 3525 mm for both growing seasons. Cumulative WW flow to the field was observed around 4920 and 5400 m$^3$, respectively, for 2018 and 2019 growing seasons, and outflow of WW was seen up to 2649 and 2931 m$^3$ correspondingly in 2018 and 2019 (Table 1). The flow of irrigation increased with the progress of time due to leaf area and willow growth development. Willow has significantly affected irrigation demands and resulted in variable WW flow and outflow between years and months.

During the growing season in 2018, the minimum aggregated loads of N and P were 89.80 and 0.66 kg·ha$^{-1}$, respectively, and in 2019 the load was increased by about 129.04 and 1.15 kg·ha$^{-1}$ for N and P, respectively. The outflow of N and P during the growing seasons is shown in Table 2.

Weather data such as temperature, rainfall, and precipitation were recorded by an onsite meteorological station located within the field area (*Vantage Pro 2, Davis Instrument*). Cropwat software (version 8.0) developed by FAO was used to measure the reference evapotranspiration (ETo) [22]. During the winter months, the temperature was ranged between −6 to −20 °C, and during the summer months, it ranged from 14 to 26 °C. The average annual rainfall and precipitation were 61.97–117.32 mm and 64.4–125.20 mm, espectively.

**Table 1.** The flow/outflow of the processed wastewater (WW) to the field. Rainwater (RW) and evapotranspiration were observed during the 2018 and 2019 growing periods.

| Month | WW (m$^3$) | RW (m$^3$) | WW + RW (m$^3$) | Outflow (m$^3$) | Evapotranspiration (mm) |
|---|---|---|---|---|---|
| **2018** | | | | | |
| May | 600 | 112 | 712 | 247 | 465 |
| June | 800 | 213 | 1013 | 413 | 600 |
| July | 1080 | 289 | 1369 | 439 | 930 |
| August | 1240 | 386 | 1626 | 696 | 930 |
| September | 1200 | 254 | 1454 | 854 | 600 |
| Total | 4920 | 1254 | 6174 | 2649 | 3525 |
| **2019** | | | | | |
| May | 520 | 80 | 600 | 135 | 465 |
| June | 1200 | 290 | 1490 | 890 | 600 |
| July | 1240 | 386 | 1626 | 696 | 930 |
| August | 1240 | 232 | 1472 | 542 | 930 |
| September | 1200 | 68 | 1268 | 668 | 600 |
| Total | 5400 | 1056 | 6456 | 2931 | 3525 |

**Table 2.** Nitrogen and phosphorus load and outflow (kg·ha$^{-1}$) during two growing periods.

| Year | N Load (kg·ha$^{-1}$) | N Outflow (kg·ha$^{-1}$) | P Load (kg·ha$^{-1}$) | P Outflow (kg·ha$^{-1}$) |
|---|---|---|---|---|
| 2018 | 89.80 | 35.66 | 0.66 | 0.33 |
| 2019 | 129.04 | 76.38 | 1.15 | 0.78 |

*2.4. Diameter Growth, Biomass Production Assessment, and Survival*

Willows with different diameters from smallest to largest (about 30 plants) were measured every year. Formula (Equation (1)) for DW (dry weight) determination was obtained through the SPSS statistic program (IBM SPSS Statistics ver. 21) based on the data set and components variables of the formula are presented in Table 3.

$$DW = a \times D^{\,b} \tag{1}$$

where *DW*—dry weight, *D*—diameter, *a* and *b*—parameters of the model obtained through statistic program, and $R^2$—coefficient of determination.

**Table 3.** The parameter's value of the model used for biomass estimation.

| Year | Parameters of the Model | | Determination |
|---|---|---|---|
| | *a* | *b* | $R^2$ |
| 2017 | 0.042 | 2.846 | 0.986 |
| 2018 | 0.105 | 2.595 | 0.983 |
| 2019 | 0.059 | 2.861 | 0.981 |

A total of 30 willow plants were randomly selected to calculate the number of shoots and oven-dried mean dry biomass (g). The annual mortality percentage % of willows were also estimated annually based on the density of 24,000 plants per ha.

*2.5. Nitrogen and Phosphorus Accumulation and Uptake (%)*

Total N and total P concentrations were estimated as N or P concentration in willow organs multiplied with a dry weight of the relevant organ and adjusted the figures for kg/ha area [23]. Moreover, the uptake percentage for total N and P in the willow organs

(leaves and stem) was evaluated using the expression used by Karnib et al. [24] and Mohsin et al. [13] as given below:

$$Total\ N\ and\ P\ uptake = \frac{N\ and\ P\ in\ plant\ biomass\ after\ phytoremediation}{N\ and\ P\ in\ the\ soil\ before\ phytoremediation} \times 100\ (\%) \quad (2)$$

### 2.6. Chemical Analysis

Before the experimental setup, the wastewater analysis was carried out. In total, three liquid samples were collected every year during June–August for a period of two years (2018–2019), presented in Table 4. The physical properties such as temperature, pH, and conductivity of processed WW were measured using an automatic multimeter (HQ2200 Portable Multimeter, Hach Lange). Chemical properties such as total nitrogen (TN), ammonia nitrogen ($NH_4$-N), nitrate nitrogen ($NO_3$-N), and total phosphorus (TP) were determined by flow injection analysis (FIA, Xylem, IO Analytical Flow Solution FS 3700, Texas, USA) according to standardize methods SFS-EN ISO 11905-1:1998 [25], SFS-EN ISO 13395:1997 [26], and SFS-EN ISO 6878:2004 [27].

**Table 4.** Physicochemical properties of processed wastewater (WW) used for irrigation during two growing seasons, 2018 and 2019 (n = 3).

| Parameter | Unit | 2018 | 2019 |
|---|---|---|---|
| Temperature | °C | 13 | 10 |
| pH | - | 7.98 | 8.09 |
| Conductivity | $\mu S \cdot cm^{-1}$ | 1150 | 865 |
| Total N | $\mu g \cdot L^{-1}$ | 2450.43 | 1340 |
| Total P | $\mu g \cdot L^{-1}$ | 266.66 | 326 |
| $NH_4$-N | $\mu g \cdot L^{-1}$ | 1583.66 | 2133.33 |
| $NO_3$-N | $\mu g \cdot L^{-1}$ | 483.33 | 266.66 |

In addition, during the end of each growing season in 2018–2019, soil samples (at 30 cm depth) from the experimental area (Outokumpu) were collected to further properties analysis at the Finnish Environment Institutes (SYKE) Joensuu. Soil samples after collection were homogenized, transferred to Petri dishes, lyophilized, and processed following the EPA method 3052 [28]. To determine the concentration of major elements (Ca, Mg, K, Al, Fe) and trace elements (Cu, Ni, Zn, Cr, As, Cd, and Pb), a soil subsample of 0.5 g (0.001 g accuracy) was digested with $HNO_3$ and HF (3:1; Suprapur) in microwave vessels in a microwave system (180 °C) for 15 min. The solution was then filtered, dissolved in 5 mL of Suprapur 0.1 M $HNO_3$, and placed in polyethylene tubes. Dilutions of ×10, ×100, and ×1000 were prepared and analyzed in an Inductively Coupled Plasma Optical Emission Spectrometer (ICP-OES, Perkin Elmer Optima 5300 DV, PerkinElmer, Ontario, Canada) to determine the contents of the elements in the soil. The results are presented as mg/kg d.w. (d.w.—dry weight). The measurements were replicated three times. Quality control was assured by analyzing a certified reference material and "blanks", according to the same procedure. Recoveries were within 10% of the certified values, depending on the individual elements. The precision, given as relative standard deviation, was in the range of 3–5%. The detection limits (LOD) of each element were calculated as Blank + 3·SD, where SD values were the standard deviations of the blank samples (n = 5). Dry matter content in soil samples was determined by drying a sample at 105 °C within 16 h. Organic matter and ashes content was determined after burning samples in a muffle furnace (F6010, Thermolyne Thermo Scientific, MA, USA) at 450 °C within 8 h. The physiochemical properties of soils are presented in Table 5.

**Table 5.** The physicochemical properties of the Outokumpu and Siikasalmi field soil (n = 3).

| Properties | Unit | Outokumpu | Siikasalmi |
|---|---|---|---|
| Dry matter | % | 98 | 95 |
| Ash | % | 93 | 90 |
| Organic matter | % | 7 | 5 |
| pH | - | 5 | 5.6 |
| N | $Mg \cdot kg^{-1}$ d.w. | 3100 | 16 |
| P | $mg \cdot kg^{-1}$ d.w. | 1949 | 4.10 |
| K | $mg \cdot kg^{-1}$ d.w. | 2068 | 52 |
| Ca | $mg \cdot kg^{-1}$ d.w. | 2524 | 290 |
| Mg | $mg \cdot kg^{-1}$ d.w. | 5016 | 42 |
| Cu | $mg \cdot kg^{-1}$ d.w. | 86 | 10.90 |
| Ni | $mg \cdot kg^{-1}$ d.w. | 18 | 8.20 |
| Zn | $mg \cdot kg^{-1}$ d.w. | 83 | 18.20 |
| Cr | $mg \cdot kg^{-1}$ d.w. | 23 | 13.3 |
| As | $mg \cdot kg^{-1}$ d.w. | 1 | 1.10 |
| Cd | $mg \cdot kg^{-1}$ d.w. | 0.10 | 0.03 |
| Pb | $mg \cdot kg^{-1}$ d.w. | 10 | 3.40 |

Harvested willows (30 plants) were separated into leaves and stems. Nitrogen in plant samples was analyzed using the Kjeldahl method by following the protocols as described by Salam et al. [10], and P was determined by standard EPA method 3052 [28]. Willow organs were rinsed thoroughly with tap water and then with deionized water. Subsequently, willow organs were dried at 70 °C for 3 days and were digested with a mixture of 3 mL of 12 mol $L^{-1}$ of hydrochloric acid (HCl) 36–38% and 1 mL of 15 mol $L^{-1}$ nitric acid (HNO₃) 69.3% in a CEM-Mars 5 microwave oven (CEM-Mars 5, CEM Corp., Matthews, NC, USA) and analyzed in an Inductively Coupled Plasma Optical Emission Spectrometer (ICP-OES, IRIS Intrepid II XSP, Thermo Fisher Scientific, MA, USA) to determine the metals concentrations in the willow tissues.

### 2.7. Statistical Analysis

Analyses of covariance (ANCOVA) were performed using the SPSS software tool (version 25.0, IBM Corporation, Armonk, NY, USA). The data were checked for normality and homogeneity of variances (data not shown). Willow diameter and biomass production were analyzed using ANCOVA with treatment as the fixed effect. Total N and P concentrations in willow organs were analyzed using ANCOVA to determine the statistical variation ($p < 0.05$) among the years using Tukey's HSD test.

### 3. Results

*3.1. Effects of Processed Wastewater (WW) Irrigation on the Willow Diameter Growth and Total Dry Biomass Production*

Willow diameter growth varied from 8 to 32 mm between 2017 and 2019 and significantly increased up to 137% (2018) and 300% (2019) as compared to the first year 2017. When willows were exposed to WW irrigation, a slight decline of around 11% in the willow diameter growth was noticed in the first year 2017, but then in 2018 and 2019, a significant annual increase of 90% and 166% were observed, respectively (Figure 3A). Results indicate that WW application has shown a positive effect on willow diameter growth.

In the case of biomass production, willow produced the largest biomass production (39.60 t·ha⁻¹) in 2019 once irrigated with WW, followed by the years 2017 and 2018. When compared with the first year (2017), biomass production was steadily increased up to 345% and 734%, respectively, in 2018 and 2019. Results have revealed that WW application has progressively increased the willow biomass production in three years, for example, 2017 (69%), 2018 (432%), and 2019 (446%) as compared to the control site (Figure 3B).

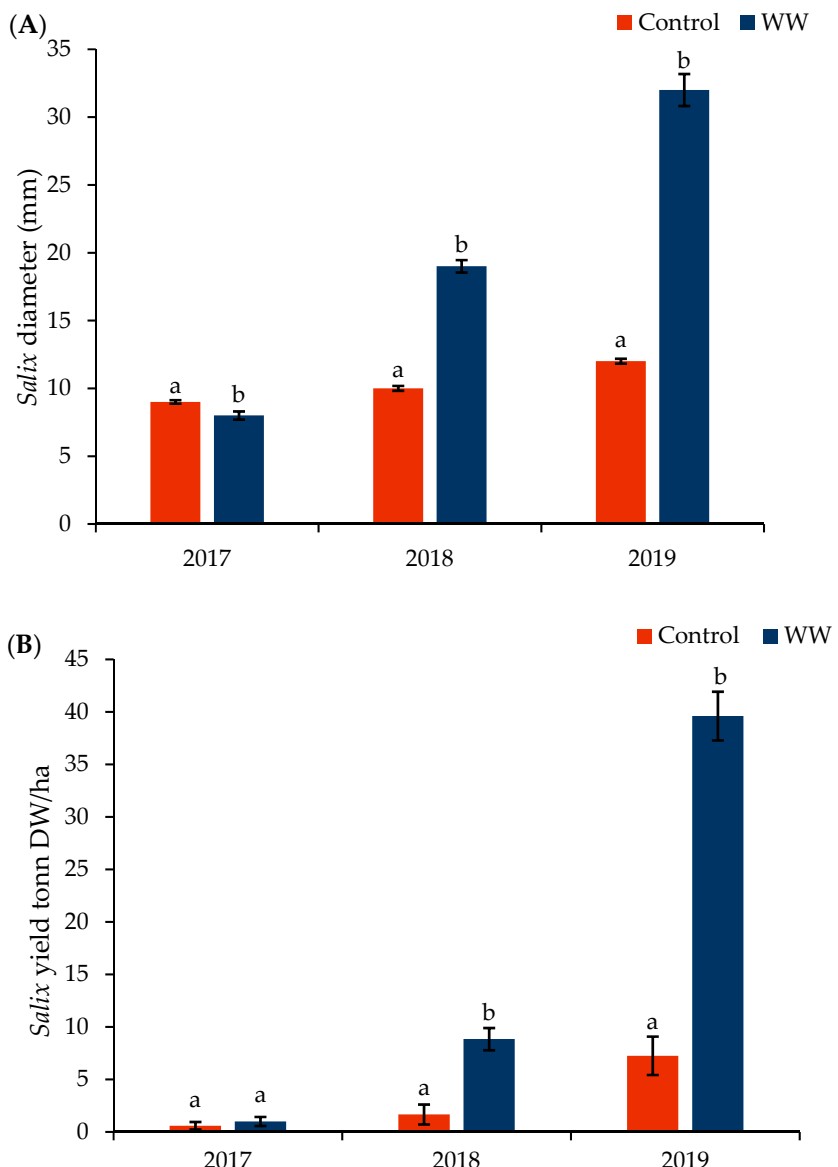

**Figure 3.** Willow diameter growth (**A**) and total dry biomass production (**B**) in three years once exposed to WW irrigation in the field (mean ± SE). Diameter (n = 30) and yield were analyzed using ANCOVA for treatment. Different small letters represent the significant difference between treatments at $p < 0.05$ using Tukey's HSD test.

Moreover, the mean dry weight of willow in different years was increased up to 705% (2018) and 2914% (2019) compared to the first year (2017). No substantial differences were observed for the numbers of shoots during the growing seasons. Though, willow exhibited the highest mortality rate (12%) in the final growth year. When compared with the control site, we observed an increase in dry weight during the growth years up to 42% (2017), 299% (2018), and 62% (2019) once willow was exposed to WW irrigations (Table 6). Moreover, WW irrigation has shown a reduction in willow mortality rate throughout the growing periods, whereas a gradual increase in mortality rate was noticed for the willows grown in the control site (Table 6).

**Table 6.** Estimation of dry weight and number of shoots (mean ± SE, n = 30), and mortality rate (density-based) of willow under WW irrigation in different years.

| Property | 2017 | | 2018 | | 2019 | |
|---|---|---|---|---|---|---|
| | Control | WW | Control | WW | Control | WW |
| Dry weight (g) | 39.32 ± 20.21 | 55.41 ± 13.49 | 111 ± 48.98 | 443 ± 146 | 1021 ± 70.45 | 1658 ± 318 |
| Number of shoots | 3 | 2.10 | 2.1 | 1.60 | 1.6 | 1.27 |
| Mortality rate (%) | 11 | 10 | 13 | 7 | 16 | 3 |

### 3.2. Nitrogen and Phosphorus Concentration in Willow Biomass

Only WW-treated willow plants were considered for analysis of N and P in plant biomass (leaves and stem) due to their higher biomass yield compared to the control plants. In addition, willow nutrient analysis was determined only for the years 2018 and 2019 due to the limited funding availability.

Figure 4 demonstrates that willow was able to accumulate the highest amount (54 kg·ha$^{-1}$·year$^{-1}$) of N in the second year (2019) as compared to the first year (2018). Nitrogen concentration was in the range 52–54 kg·ha$^{-1}$·year$^{-1}$ while no significant differences were noticed between the years, but a comparison shows that the concentration of N was marginally increased around 3% in the second year as compared to the first year. On the other hand, P concentration was also found highest in the second year (50 kg·ha$^{-1}$·year$^{-1}$) as compared to the first year (32 kg·ha$^{-1}$·year$^{-1}$). Remarkably, P concentration increased significantly by about 56% in the second year once compared with the first year, though a significant difference was observed between the years. Nevertheless, there was a marked increase noticed for total P concentration in the second year as compared to the first year.

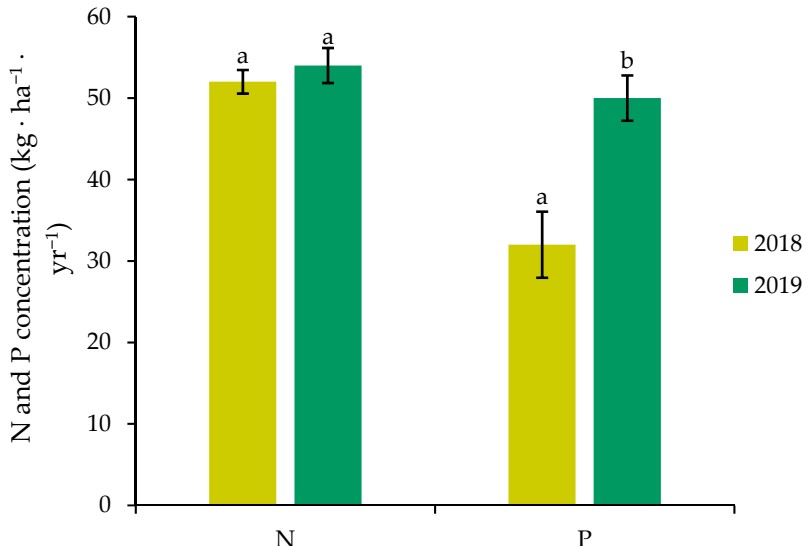

**Figure 4.** Total N and P concentrations in willow plants for two years once exposed to WW irrigation in the field (mean ± SE). N and P concentrations were analyzed using ANCOVA for years. Different small letters represent the significant difference between years at *p* < 0.05 using Tukey's HSD test.

### 3.3. Total Nitrogen and Phosphorus Accumulation Percentage

The highest N (60%) and P (50%) uptake were observed in the second year as compared to the first year. For example, in the second year, an increment of 46% and 56% was noticed for N and P, respectively, as compared to the first year (Figure 5). However, during two growing periods, nitrogen accumulation in willow was observed higher than P. Significant difference (*p* < 0.05) was seen between the years (2018 and 2019) for both N and P uptake.

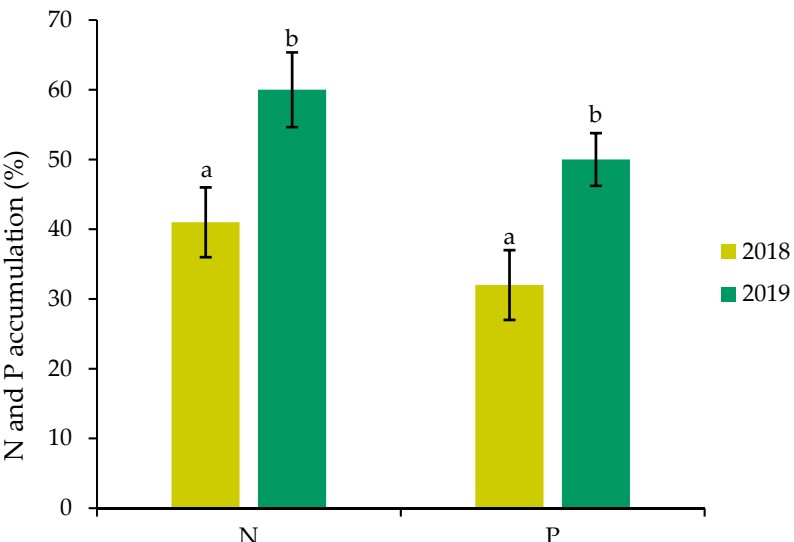

**Figure 5.** Total N and P accumulation by willow for two years once exposed to WW irrigation in the field (Mean ± SE). N and P accumulation (%) were analyzed using ANCOVA for years. Different small letters represent the significant difference between years at *p* < 0.05 using Tukey's HSD test.

### 3.4. Nutrient and Metal Concentrations in the Soil Treated with Processed Wastewater (WW)

Soil analyses were carried out for only WW-treated soil in the Outokumpu study area. We analyzed nutrient (N, P, Mg, Ca, K, Fe, and Al) and heavy metals (Cu, Ni, Zn, Cr, Cd, Pb, and As) concentrations in WW-treated soil at the end of the growing season in 2018 and 2019. Results demonstrated that WW-treated soil contained large concentrations of N (401 mg·kg$^{-1}$) and P (265 mg·kg$^{-1}$) including other nutrients, though very low heavy metals concentration (0.002–3.04 mg·kg$^{-1}$) was detected in soils for Cu, Ni, Zn, Cr, Cd, Pb, and As. Overall, the pattern of total nutrient concentration in soil were observed as N > P > Fe > Al > Mg > Ca (Figure 6). Moreover, WW has nutritious properties since it might be used to irrigate crops to reduce the dependence on freshwater usage.

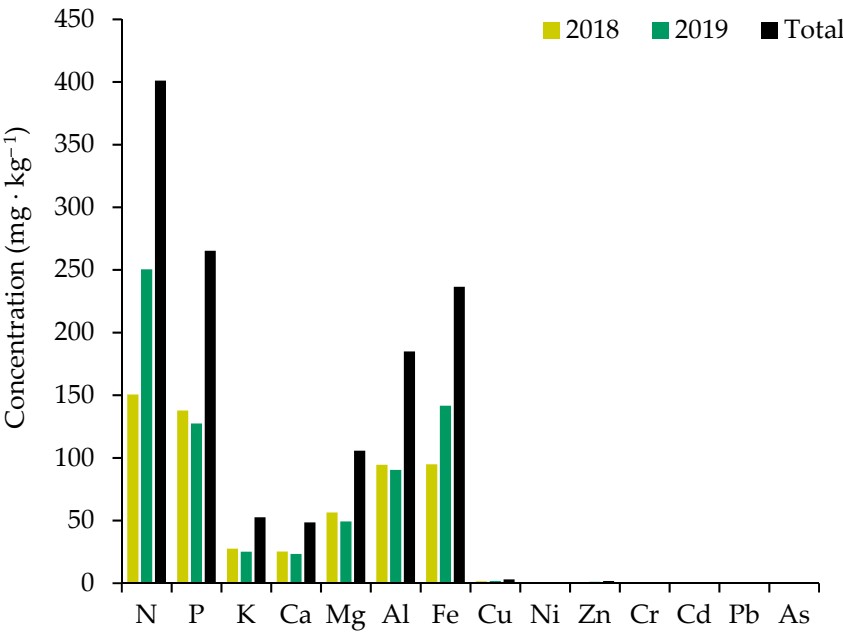

**Figure 6.** Nutrient and heavy metals concentrations in WW-treated soil (Outokumpu area) at the end of growth seasons (n = 3; depth 0–30 cm).

## 4. Discussion

The main purpose of this study was to investigate the effect of processed wastewater (WW) irrigation on the biomass production and diameter growth of willow and uptake of total N and P into willow tissues under field conditions. An additional goal of the willow plantation near the Outokumpu wastewater treatment plant was to control the high discharge of nutrients and prevent eutrophication in the surrounding waters. Globally, very few studies have been published considering the phytoremediation efficiency of short-rotation woody plants under WW irrigation. Nevertheless, our results showed that the studied willow variety had produced relatively high biomass yields compared to control soil (without WW). The increase in biomass production might be due to high nutrients uptake by willow from WW needed for optimal growth development, although dense planting has also been reported to increase biomass yields in multiple years [29]. Moffat et al. [30] investigated the effect of sewage sludge and wastewater irrigation on the biomass production of two poplar varieties (*Populus trichocarpa×P. deltoides Beaupré, and P. trichocarpa Trichobel*) and reported yields up to 8 ton· ha$^{-1}$·year$^{-1}$ dry biomass in three years, which are noticeably lower than those achieved in this study, for example, the highest one was observed 36 ton· ha$^{-1}$·year$^{-1}$. In a previous study, stem biomass of two willow varieties such as *Salix viminalis* and *Salix discolor* were found significantly greater when receiving wastewater treatment, resulting in productivity exceeding that of control plants by five to seven times [31]. Similarly, Kowalik and Randerson [32] have also reported a considerable yield for *Salix amygdalina* (14.4 ton· ha$^{-1}$·year$^{-1}$), *Salix viminalis* (8.7 ton· ha$^{-1}$·year$^{-1}$), and *Salix americana* (7.1 ton· ha$^{-1}$·year$^{-1}$) under municipal wastewater irrigation for two-year rotation period in Wrocław, Poland.

Short-rotation coppice is a highly compatible practice and can be environmentally sustainable in terms of land treatment as it reduces public health risks and increases nutrient recycling [33]. Studies suggest that phytoremediation by specific *Salix* clones can be a suitable option in technical replenishment approaches in soil remediation. *Salix* growth in contaminated soil is generally regulated by the soil conditions and the presence of other plants or weather conditions [10,13,34]. The municipal wastewater purification potential of different willow species has been investigated in Poland, Sweden, Estonia, and Denmark. In Sweden, approximately five municipalities have been using willow as a vegetation filter as a complement to commercial wastewater treatment methods, especially for nutrient and heavy metals removal [35]. In addition, in a field trial in the uplands of Mid-Wales, U.K., positive effects of treated sewage sludge and wastewater on willow coppice yield have been documented by Heaton et al. [36]. In the rural areas of the Nordic countries, many homes are dependent on onsite systems. In addition, in Finland, most of the countryside homes are connected to sewerage systems to municipal level wastewater treatment systems. Hence, there is a need for a vigorous and compact onsite treatment system that can remain effective during cold climates and efficiently remove and recover nutrients [5].

In the current study, willow showed 41–60% N and 32–50% P removal when subjected to WW irrigation containing 80–129 kg N ha$^{-1}$ year$^{-1}$ and 0.66–1.15 kg P·ha$^{-1}$·year$^{-1}$. The results are in line with the findings of Holm and Heinsoo [37], who set up a field trial in Estonia and has reported that willow was able to retain up to 58% N and 70% of total P under the application of WW, which contained 29 kg N·ha$^{-1}$·year$^{-1}$ and 4 kg P·ha$^{-1}$·year$^{-1}$. Similarly, Amofah et al. [5] observed a significant accumulation of total N and P in two willow clones Karin (((*Salix schwerinii* × *Salix viminalis*) × *Salix viminalis*) × *Salix burjatica*) (212 and 28 kg·ha$^{-1}$) and Gudrun (*Salix dasyclados*) (212 and 27 kg·ha$^{-1}$) respectively under WW irrigation for three years.

Noticeably, in the current study, total N accumulation (41–60%) was highest than P (32–50%), which indicates that N accumulation potential is higher than P because the willow demands more N than P as a nutritive element and absorb N (around 5–10 times) faster than P [38]. However, it is expected that the longer the plants can remain in situ and absorb nutrients, the greater the number of nutrients that will be extracted [13].

Several factors depend on the nutrient accumulation in plants, such as experimental time, genetic differences in plants, total nutrient concentration in soil [10], physiochemical properties of the soil, organic matter content, and soil pH [39]. Furthermore, WW has shown positive effects on growth height, dry biomass production, and nutrient accumulation in a willow cultivar *Salix schwerinii* under greenhouse [10]. Similarly, another study by Salam et al. [11] has reported an increase in growth height, dry biomass, shoot diameter, and nutrient accumulation in a willow cultivar Klara once exposed to WW irrigation under greenhouse. These greenhouse investigations support the finding from our field trial in terms of diameter growth improvement, increase in biomass production, and significant N and P accumulation in *Salix schwerinii* under WW irrigation. The proportion of nutrients recovered varies greatly and is contingent on the nutrient concentration in wastewater, nutrient loading, the timing of application and harvesting intervals, the ability of plant species for nutrient uptake, and accumulation in aboveground biomass [40]. We observed high nutrient and very low metal concentration in soils after growth season (Figure 4), which represents that WW used in the study has nutritious properties and could be used for agriculture purposes.

It has been reported that when soil is irrigated with metals enriched wastewater or WW harbors, a diverse community of microbes is resilient to toxic metals and contributes to the remediation of wastewater [41]. Soil microbes are located in the close vicinity of plant roots, and most of them enter root cortical cells to act as endophytes that facilitate plant growth development by using various mechanisms include phosphate solubilization, growth hormone synthesis, zinc mobilization, induction of stress tolerance, ACC (1-aminocyclopropane-1-carboxylic acid) deaminase activity, siderophore production, nitrogen fixation and biocontrol activity [42]. Furthermore, the root exudates released in plant root environs consist of different organic acids include oxalate, acetate, fumarate succinate, lactate, citrate, and malate, which endure in the soil as dissociated anions, amino acids, and sugars, and some secondary metabolites, for example, isoprenoids, flavonoids, and alkaloids [43]. These exudates facilitate an affirmative environment for soil microbes by donating their nutrients, which resultantly offer plant growth-promoting hormones and nutrients, produce systemic resistance against abiotic and biotic stress that ameliorate roots architecture and plant growth [44]. However, microbes and plant interaction are generally contingent on a number of factors such as soil type as well as pollutant concentration and microbial diversity in polluted soils [45]

Presently, N leaching is a major threat to the Baltic Sea; meanwhile, agricultural activities account for 70–90% of the total diffuse N load. The Baltic Sea Action Plan demonstrates that a drastic reduction in nutrient load to the Baltic Sea is required, and different measures need to be implemented as nutrients reductions have particular importance for the areas that are at risk due to high nutrients leaching [46]. In this context, short-rotation crops such as willows with efficient N and P uptake could play a complementary role in fulfilling nutrient reduction targets along with high biomass production and recovery of valuable elements to strengthen the circular economy drives [5].

## 5. Conclusions

Phytoremediation by *Salix schwerinii* offers an eco-friendly mechanism for conventional remediation of nutrients enriched wastewater as *Salix* has a deep-root system and provides high biomass yields. Based on a density of 24,000 plants per ha, willow showed a total of 20% mortality rate throughout the growing periods (2017–2019) under WW irrigation compared to the control site (39%). Willow has shown an increase in the biomass yield by about 64–444% between 2017 and 2019 and stems diameter growth around 90–166% during 2018–2019 under processed wastewater (WW) irrigation as compared to control site (without WW application). *Salix* has also exhibited an ability to accumulate a total amount of N (23%) and P (50%) in the second growing year (2019) as compared to the first year (2018). Moreover, the results of this study would contribute to the effective management of different wastewaters and wetlands to balance the nutrient level. Though, the willow

variety should be selected carefully, as survival efficiency and biomass production potential may vary between varieties. Before and after the growth experiment, very low metal concentration was found in the soil irrigated with WW. Hence, WW can be a source of fertilizer as it has a significant contribution of N, P, and organic matter, can save farmers money on fertilizers, and reduce pressure to use freshwater resources in society. Additional research is needed to examine the complementary effects of organic (sewage sludges) and chemical fertilizers (NPK) on the yield of different short-rotation bioenergy crops and the microbial biomass composition of agricultural soils.

**Author Contributions:** Conceptualization, A.V.; Funding acquisition, A.P.; Investigation, E.K., N.E., and A.V.; Supervision, S.K. and A.P.; Writing—original draft, M.M.; Writing—review and editing, E.K., M.M.A.S., N.N., and E.W. All authors have read and agreed to the published version of the manuscript.

**Funding:** The study has been funded by the Karelia CBC program, the European Union, the Russian Federation, and the Republic of Finland. The project entitled "Reaching congenial region through the valorization of municipal and industrial wastewaters and sludge" (project no: KA4020). The first author (M.M.) would like to thank Niemi Foundation (grant no. 20190036) for supporting this research work.

**Institutional Review Board Statement:** Not applicable.

**Informed Consent Statement:** Not applicable.

**Data Availability Statement:** Data available on request from the corresponding author.

**Conflicts of Interest:** The authors declare no conflict of interest. The funders had no role in the design of the study, in the collection, analyses, or interpretation of data, in the writing of the manuscript, or in the decision to publish the results.

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
