# Peer review of "Biomass Production and Removal of Nitrogen and Phosphorus from Processed Municipal Wastewater by Salix schwerinii: A Field Trial"

_water, doi:10.3390/w13162298_

Round 1
Reviewer 1 Report
Water-1318408. Biomasss prodn Potl and PhytoRem of N & P irrigated with WW – reviewer comments.
This paper describes a field trial in Finland to assess the effectiveness of a young willow plantation for biofiltration of municipal WW, in the context of the northern European climate. Such field scale systems are already widely used around the World with the aim of reducing the influx of nutrients (N & P) into surface water bodies causing eutrophication, as well as enhancing the production of woody biomass as a source of bioenergy and their adoption by municipal authorities should be applauded. Although not offering a full review of the subject, the paper provides adequate context and background in the list of references, with a focus on Nordic countries as a model for willow growth under seasonally harsh environmental conditions. The text is well written (the few minor errors/unclarities are listed below). This report provides a useful addition to the growing literature on biofiltration of WW by SRC which, as the authors make clear, is of vital importance for countries bordering the Baltic sea, The following suggestions are made with the am of improving the context of this recent work, which follows from pioneering studies undertaken in the 1990s (mainly in Poland and Sweden) at a time, perhaps, before the concept of the bioeconomy became mainstream.
The final para of Introduction (line 92-3) “very little attention has been paid concerning the effects of WW irrigation on short-rotation willow”, implies that this recent study is a novel concept – “We hypothesize that willow plantations would accumulate the highest amount of nutrients from the WW and produce high biomass yield under WW irrigation.” I am sure this is not the authors’ intention since, in the Discussion, they reference older work from Poland, Sweden, Estonia & Denmark (e.g. Perttu & Kowalik 1997). However, in the interests of balance and context, I would encourage the authors to include other “historical” references to the irrigation of SRC willow with WW and sewage sludge (as distinct from WW treatment in constructed wetlands). For example:
Kowalik, P. and Randerson, P. 1994. Nitrogen and phosphorus removal by willow stands irrigated with municipal waste water - a review of the Polish experience. Biomass and Bioenergy 6(1-2), pp. 133-139.
Heaton, R. J.et al. 2001. The influence of fertilisation on the yield of short rotation willow coppice in the uplands of mid-Wales. Aspects of Applied Biology 65, pp. 77-82.
Hence Wales, UK, could also be added to the list of countries where pioneering trials of sewage sludge and WW application to willow coppice were under way (line 349) – or do the authors wish to focus on the Baltic?
Results:
Line 241. …but then in 2018…
Lines 246 and 257. Delete “was” – redundant word.
Line 262. Add “was” …once willow was exposed to…
Lines 282-3. The meaning is not clear and needs to be re-written. Do you mean …Remarkably, P concentration increased significantly by about 56% in the 2nd year?
Line 284-5. “a much decline” should be …a marked decline between… I cannot relate this statement to the pattern shown in Fig 2. Please clarify.
Line 292. Percentage
Line 293. …P (nearly 50%) uptake… - is this what was intended?
Lines 314/5. It appears from Fig 4 to be N>P>Fe>Al>K>Mg, rather than <<<<
Discussion:
Line 330. …has produced relatively high…
Line 344. …studies confirm that.. – or …studies suggest that…
Line 355. To avoid “efficient” used twice, I suggest …remain effective … and efficiently remove..
Line 393. …N leaching is the main threat… - or …is a major threat…
Author Response
This paper describes a field trial in Finland to assess the effectiveness of a young willow plantation for biofiltration of municipal WW, in the context of the northern European climate. Such field scale systems are already widely used around the World with the aim of reducing the influx of nutrients (N & P) into surface water bodies causing eutrophication, as well as enhancing the production of woody biomass as a source of bioenergy and their adoption by municipal authorities should be applauded. Although not offering a full review of the subject, the paper provides adequate context and background in the list of references, with a focus on Nordic countries as a model for willow growth under seasonally harsh environmental conditions. The text is well written (the few minor errors/unclarities are listed below). This report provides a useful addition to the growing literature on biofiltration of WW by SRC which, as the authors make clear, is of vital importance for countries bordering the Baltic sea, The following suggestions are made with the am of improving the context of this recent work, which follows from pioneering studies undertaken in the 1990s (mainly in Poland and Sweden) at a time, perhaps, before the concept of the bioeconomy became mainstream.
The final para of Introduction (line 92-3) “very little attention has been paid concerning the effects of WW irrigation on short-rotation willow”, implies that this recent study is a novel concept – “We hypothesize that willow plantations would accumulate the highest amount of nutrients from the WW and produce high biomass yield under WW irrigation.” I am sure this is not the authors’ intention since, in the Discussion, they reference older work from Poland, Sweden, Estonia & Denmark (e.g. Perttu & Kowalik 1997). However, in the interests of balance and context, I would encourage the authors to include other “historical” references to the irrigation of SRC willow with WW and sewage sludge (as distinct from WW treatment in constructed wetlands). For example:
Kowalik, P. and Randerson, P. 1994. Nitrogen and phosphorus removal by willow stands irrigated with municipal waste water - a review of the Polish experience. Biomass and Bioenergy 6(1-2), pp. 133-139.
Response: Thank you very much for your valuable comments and your positive review. We have referred to your comments and marked corrections throughout the manuscript so that they can be quickly followed using the “track changes” option. Below we provide detailed answers to your comments. We have added some discussion concerning biomass production in the manuscript from the suggested article.
Heaton, R. J.et al. 2001. The influence of fertilisation on the yield of short rotation willow coppice in the uplands of mid-Wales. Aspects of Applied Biology 65, pp. 77-82. Hence Wales, UK, could also be added to the list of countries where pioneering trials of sewage sludge and WW application to willow coppice were under way (line 349) – or do the authors wish to focus on the Baltic?
Response: Dear reviewer, thank you for your suggestion. We have added the UK to the list of countries where willow coppice investigation under sewage sludge and WW is underway. One review suggested reducing discussion length while another suggested adding more microbial aspects in phytoremediation; therefore, it would be challenging to explain in detail UK trials in the manuscript.
Results:
Line 241. …but then in 2018…
Response: “the” replaced with “then”
Lines 246 and 257. Delete “was” – redundant word.
Response: “was” deleted
Line 262. Add “was” …once willow was exposed to…
Response: “was” added
Lines 282-3. The meaning is not clear and needs to be re-written. Do you mean …Remarkably, P concentration increased significantly by about 56% in the 2nd year?
Response: “about 56 % P concentration was 282 increased in the second year once compared with the first year” has been revised as “P concentration increased significantly by about 56% in the second year”
Line 284-5. “a much decline” should be …a marked decline between… I cannot relate this statement to the pattern shown in Fig 2. Please clarify.
Response: The sentence has been revised as “Nevertheless, there was a marked increase noticed for total P concentration in the second year as compared to the first year.”
Line 292. Percentage
Response: Revised
Line 293. …P (nearly 50%) uptake… - is this what was intended?
Response: This uptake was not indented that much. We have revised the sentence as “The highest N (60 %), and P (50 %) uptake was observed in the second year as compared to the first year”. “nearly” has been removed from the sentence.
Lines 314/5. It appears from Fig 4 to be N>P>Fe>Al>K>Mg, rather than <<<<
Response: It has been revised.
Discussion:
Line 330. …has produced relatively high…
Response: It has been revised.
Line 344. …studies confirm that.. – or …studies suggest that…
Response: It has been revised as “studies suggest that” in the sentence.
Line 355. To avoid “efficient” used twice, I suggest …remain effective … and efficiently remove.
Response: “Efficient” has been replaced with “effective”.
Line 393. …N leaching is the main threat… - or …is a major threat…
Response: “main” has been replaced with “major” in the sentence.
Thank you very much once again for your essential comments and support while improving the scientific value of this manuscript. We hope that implemented corrections will be gratifying for you in making a favourable decision about it.
Reviewer 2 Report
In the present papers, the authors investigated the effect of municipal wastewater on the growth diameter and total dry biomass of willow over three-year period. The also assessed total N and P accumulation by willow when irrigated with WW.
General comments
The paper is well-structured and easy to read. Introduction provides comprehensive state of the art and is focused on the topic. Material and methods are adequately detailed. Results section provides the relevant findings of the experiments, and they are contrasted against literature.
Specific comments
Line 40: “achieving food security”. Why food security sector would be positively affected by municipal wastewater treatment?
Line 45: “of 3400 tons of P and 74,000 tons of N”. When talking about eutrophication from N and P, it is usual refer first to N and then to P as the authors did at the beginning of the introduction. Thus, the authors should be consistent with this throughout the paper.
Line 123: a full stop is missing after “installed to the wastewater treatment plant”
Legend of Table 1: “Rainwater (RW) and evapotranspiration were observed during the 2017 and 2018 growing periods.” In the table 2018 and 2019 are indicated, please clarify.
Line 168: Details of SPSS package should be removed from Line 172 and put after “SPSS statistic program” in this line.
Line 387: add reference at the end of sentence (after remediation of wastewater).
Author Response
Specific comments
Line 40: “achieving food security”. Why food security sector would be positively affected by municipal wastewater treatment?
Response: Thank you for pointing this out. Currently, freshwater resources are at risk globally, and soon, it would be challenging to fulfil the water demand for drinking and crop irrigation. For the agricultural sector, the reuse of treated wastewater could be a viable solution for crop irrigation by reducing pressure on freshwater resources to face a challenging situation in the future. So, freshwater used for agriculture can be safe for drinking purposes instead of agriculture uses.
Line 45: “of 3400 tons of P and 74,000 tons of N”. When talking about eutrophication from N and P, it is usual refer first to N and then to P as the authors did at the beginning of the introduction. Thus, the authors should be consistent with this throughout the paper.
Response: Thank you for your careful reading. It has been revised throughout the paper.
Line 123: a full stop is missing after “installed to the wastewater treatment plant”
Response. a full stop is added.
Legend of Table 1: “Rainwater (RW) and evapotranspiration were observed during the 2017 and 2018 growing periods.” In the table 2018 and 2019 are indicated, please clarify.
Response. Caption in Table 1, during the 2017 and 2018 growing periods has been revised as during the 2018 and 2019 growing periods
Line 168: Details of SPSS package should be removed from Line 172 and put after “SPSS statistic program” in this line.
Response: It has been revised.
Line 387: add reference at the end of sentence (after remediation of wastewater).
Response: The references have been cited.
Thank you very much once again for your essential comments and support while improving the scientific value of this manuscript. We hope that implemented corrections will be gratifying for you in making a favourable decision about it.
Reviewer 3 Report
This is an interesting manuscript however it has to be improved
1) the title is very long. I understand that you want to be precise but it has to be shortened eg Removal of Nitrogen and Phosphorus from Processed Municipal Wastewaterby Salix schwerinii: A Field Trial
2) some English language proofreading is needed. eg replace
as well as the nutrients uptake capability (mainly N and P) of a willow variety 19
(Salix schwerinii).
with
as well as nutrients uptake (mainly N and P) by a willow variety 19
(Salix schwerinii).
replace
by two willow cultivars 59
include Salix schwerinii and Klara
with
by two willow cultivars 59 that
include Salix schwerinii and (give also the latin name)
this is just an example many more corrections are needed
3) there are some carelessness mistakes eg it should be of 3,400 tn of P and 74,000 tn. also CH4 etc... should be with subscript.
4) I dont understand how oxidation processes may give CO2 please elaborate
5) I dont understand if you use the willow cultivation instead of nitrification/denitrification. as far as I know, this process takes place in anoxic/aerobic parts of the secondary treatment and you will use treated wastewater anyway? so your method does not really replace nitrification/denitrification? please elaborate
6) the introduction is very long. I suggest to shorten considerably. also make clearer what is the novelty of your research since this method has been tried before, in the conclusion of your introduction
7) it would be good to have a map (eg google map) of the experimental area
8) some more data for the WWTP are needed eg average flow also, design charecteristics eg BOD, type of WWTP (eg suspended biomass, tertiary treatment?, chlorination? etc). is there some protocol for the planting of willows eg from some previous work?
9) the materials and methods should be rewritten because they are quite confusing. you write that N, P and metals were measured by ICP-OES whereas it is clear that N, P were measured with other methods. give reference for the P method. give LOD/LOQ for all the elements measured. you have to first describe the sample preparation (eg pulverization) and then the analytical method. you have to give manufacturer, city country of origin for all apparatuses used (eg furnace, pulverizer, ICP-OES, glassware, centrifuge etc)
10) you do not give some very important information eg Weather data, rainfall, and precipitation were recorded by an on–site meteorological 160
station located within the field area. what does this mean? what station was this? what is Cropwat software (manufacturer) and what it measures exactly. you state WW irrigation to willow was based on crop water requirements but this is not explained further-how exactly was this measured? you never described how Conductivity, NH4-N, NO3-N, Dry matter, Ash, Organic matter were measured
11) if you wanted to measure the differences in N and P within the years you should use a repeated measures ANOVA not an one-way ANOVA
12) I believe that a diagram that shows the irrigation scheme would aid very much the understanding of the manuscript
13) in the graphs if there are different letters on each bar that means that there is a statistically significant difference. if there are asterisks this means that there is a statistically significant difference and *** means P<0.001. So you should not have both
14) please rearrange the discussion to 1/3 of present length and give some simple, concise explanations. do not give numbers-these can be seen in the graphs. for such a long manuscript 40-50 references are essential
Author Response
This is an interesting manuscript however it has to be improved
Thank you very much for your positive feedback on our manuscript. We have referred to your comments and marked corrections throughout the manuscript so that they can be easily followed using the “track changes” option. Below we provide detailed answers to your comments.
1) the title is very long. I understand that you want to be precise but it has to be shortened eg Removal of Nitrogen and Phosphorus from Processed Municipal Wastewaterby Salix schwerinii: A Field Trial
Response: Thank you for the suggestion. As the study also focuses on biomass production; therefore, the manuscript title has been revised as Biomass Production and Removal of Nitrogen and Phosphorus from Processed Municipal Wastewater by Salix schwerinii: A Field Trial
2) some English language proofreading is needed.
Thank you very much for your careful reading. We corrected it according to your suggestion and checked the manuscript.
eg replace as well as the nutrients uptake capability (mainly N and P) of a willow variety 19
(Salix schwerinii). with as well as nutrients uptake (mainly N and P) by a willow variety 19
(Salix schwerinii).
Response. “Of” has been replaced with “by”.
Replace by two willow cultivars 59
include Salix schwerinii and Klara with by two willow cultivars 59 that
include Salix schwerinii and (give also the latin name)
this is just an example many more corrections are needed
Response: It has been revised as Klara (Salix viminalis x Salix schwerinii x Salix dasyclados). Similar corrections have been made throughout the manuscript.
3) there are some carelessness mistakes eg it should be of 3,400 tn of P and 74,000 tn. also CH4 etc... should be with subscript.
Response: Thank you very much for pointing this out. 3400 tn of P and 74,000, CH4, N2O, CO2 have been revised as 3,400 tn of P and 74,000, CH4, N2O, CO2 in the manuscript. And similar mistakes have been revised throughout the manuscript.
4) I dont understand how oxidation processes may give CO2 please elaborate—5) I dont understand if you use the willow cultivation instead of nitrification/denitrification. as far as I know, this process takes place in anoxic/aerobic parts of the secondary treatment and you will use treated wastewater anyway? so your method does not really replace nitrification/denitrification? please elaborate
Response: Dear reviewer, our method will not replace the conventional method; our main intention was to reuse processed wastewater from treatment plants that still contain N and P that can be used in a circular economy approach to irrigating crops to avoid freshwater usage. Willow plantations would help balance the discharged gases in the environment if grown in the surrounding treatment plant. Due to conflicting statements in the introduction (Conventional wastewater treatment methods have been deployed to reduce the nutrient load, but they are expensive and might have climate change impacts by emitting various greenhouse gases to the atmosphere, for example, methane (CH4) from anaerobic processes, nitrous oxide (N2O) from nitrification/denitrification processes and carbon dioxide (CO2) from aerobic (oxidation) processes), we have decided to remove that statement since you also suggested to reduce the introduction.
6) the introduction is very long. I suggest to shorten considerably. also make clearer what is the novelty of your research since this method has been tried before, in the conclusion of your introduction
Response: Dear Reviewer, thank you very much for your suggestion. However, we would like to point out that our introduction is 887 words, which is within the average length of the Introduction section. We tried to remove unnecessary spaces that could make the introduction seem too long. In order to maintain the length of the Introduction, we would like to mention that other Reviewers agree with the current length of the Introduction, and some have suggested adding information about microbe’s role in phytoremediation, etc. It would be very tough to exclude some paragraphs. Our study focused on the circular economy approach as the reuse of nutrient-enriched processed water could fulfil the irrigation requirements of crops by reducing pressure on freshwater usage and chemical fertilizers. We have revised the novelty in the last paragraph of the introduction.
7) it would be good to have a map (eg google map) of the experimental area
Response: Thank you for this advice. The maps from the experiential areas have been added to the manuscript.
8) some more data for the WWTP are needed eg average flow also, design charecteristics eg BOD, type of WWTP (eg suspended biomass, tertiary treatment?, chlorination? etc). is there some protocol for the planting of willows eg from some previous work?
Response: More information regarding BOD, WWTP treatment has been added in the material and methods of introduction. Sorry for the mistake, the average flow of WWTP is 3,750 m3 / day. In the manuscript, we wrote a full 3750m3/day, which has been replaced with “average”. A detailed explanation has been added in the materials and method part.
9) the materials and methods should be rewritten because they are quite confusing. you write that N, P and metals were measured by ICP-OES whereas it is clear that N, P were measured with other methods. give reference for the P method. give LOD/LOQ for all the elements measured. you have to first describe the sample preparation (eg pulverization) and then the analytical method. you have to give manufacturer, city country of origin for all apparatuses used (eg furnace, pulverizer, ICP-OES, glassware, centrifuge etc)
Response: Thank you very much for pointing this out. We tried to improve these issues. A detailed explanation about the chemical analysis of N, P, and metals has been revised and added in the manuscript.
10) you do not give some very important information eg Weather data, rainfall, and precipitation were recorded by an on–site meteorological 160
station located within the field area. what does this mean? what station was this? what is Cropwat software (manufacturer) and what it measures exactly. you state WW irrigation to willow was based on crop water requirements but this is not explained further-how exactly was this measured? you never described how Conductivity, NH4-N, NO3-N, Dry matter, Ash, Organic matter were measured
Response: Dear Reviewer, an on-site metrological instrument (Vantage Pro2, Davis) was installed in the field to estimate rainfall, temperate, and precipitation. The information has been added to the manuscript. CROPWAT is a decision support tool developed by the Land and Water Development Division of FAO. CROPWAT 8.0 for Windows is a computer program generally used to calculate crop water requirements, evapotranspiration, and irrigation requirements based on soil, climate, and crop data. In addition, the method regarding Conductivity, NH4-N, NO3-N, Dry matter, Ash, Organic matter has been added to the manuscript. In our study, willow irrigation at a rate of 10mm day-1 was based on Swedish model by Dimitriou and Aronsson, 2004 (ref cited has been cited in the manuscript)
11) if you wanted to measure the differences in N and P within the years you should use a repeated measures ANOVA not an one-way ANOVA
Response: Dear reviewer, we apologize for the mistake; it was an analysis of covariance (ANCOVA) with the Tukey test, not a one-way ANOVA. The analysis in the statistical section has been revised.
12) I believe that a diagram that shows the irrigation scheme would aid very much the understanding of the manuscript
Response: Thank you for this suggestion. We have added the diagram in the manuscript (Figure 2). We hope that now its more clear.
13) in the graphs if there are different letters on each bar that means that there is a statistically significant difference. if there are asterisks this means that there is a statistically significant difference and *** means P<0.001. So you should not have both
Response: Thank you once again for your careful reading. The asterisks indication has been removed from all the graphs.
14) please rearrange the discussion to 1/3 of present length and give some simple, concise explanations. do not give numbers-these can be seen in the graphs. for such a long manuscript 40-50 references are essential.
Response: Dear Reviewer, other reviewers suggested adding more discussion about results compared to other studies. Therefore, it would be very difficult to reduce the discussion length. The numbers given in the Discussion support our findings and we do hope that they were quoted correctly in the Discussion and not in the Results section. We do hope that you share our opinion and agree to leave this data in the Discussion section. Thank you for your understanding. We tried to satisfy the requirements of all Reviewers and provide a reasonable balance between the length of our paper and additional information. As a result of the introduced amendments, the list of our references has expanded: there are 46 references in the current version.
Thank you very much once again for your essential comments and support while improving the scientific value of this manuscript. We hope that implemented corrections will be gratifying for you in making a favourable decision about it.
Reviewer 4 Report
Mohsin et al. carried a a four–year field trial to investigate the effect of the processed wastewater on biomass production as well as the nutrients uptake capability (mainly N and P) of the narrow-leaf willow (Salix schwerinii).
Both the title and the abstract are relevant and concise.
The paper is of general good quality and the length of the trial is an important addition, therefore the work performed will bring a substantial contribution to the current literature.
The Introduction however lacks the microbial aspect.
Line 54: " Phytoremediation is an eco-friendly and cost-effective method for" More information regarding phytoremediation [1] and especially regarding phytomicrobiome and it's biotechnological potential should be added [2].
Materials and Methods
Line 102: GIS software to establish coordinates should be stated
Line 200-202: Briefly state the method not only cite it
Line 386-390: "a diverse community of microbes resilient to toxic metals and contribute to the remediation of wastewater. Soil microbes are located in the close vicinity of plant roots and most of them enter root cortical cells to act as endophytes which facilitate plant growth" This, as previously mentioned, should be further detailed as it is an important and relevant chapter of the presented work.
References
- Peuke AD, Rennenberg H. Phytoremediation. EMBO Rep. 2005;6(6):497-501. doi:10.1038/sj.embor.7400445
- FENDRIHAN S, POP CE. Biotechnological potential of
plant associated microorganisms. Rom Biotechnol Lett. 2021; 26(3): 2700-2706. DOI: 10.25083/rbl/26.3/2700-2706
Author Response
Mohsin et al. carried a a four–year field trial to investigate the effect of the processed wastewater on biomass production as well as the nutrients uptake capability (mainly N and P) of the narrow-leaf willow (Salix schwerinii).
Both the title and the abstract are relevant and concise.
The paper is of general good quality and the length of the trial is an important addition, therefore the work performed will bring a substantial contribution to the current literature.
Response: Thank you very much for your positive review. We have referred to your comments and marked corrections throughout the manuscript so that they can be easily followed using the “track changes” option. Below we provide detailed answers to your comments.
The Introduction however lacks the microbial aspect.
Line 54: " Phytoremediation is an eco-friendly and cost-effective method for" More information regarding phytoremediation [1] and especially regarding phytomicrobiome and it's biotechnological potential should be added [2].
Response: Thank you very much for this suggestion. We tried to add more information about phytoremediation from ref (1) and Phytomicrobiome ref (2) (lines 64-72). Other Reviewers already commented that the Introduction is long that’s why we tried to wind up the Introduction as it is in the present length. We tried to satisfy the requirements of all Reviewers and provide a reasonable balance between the length of our paper and additional information.
Materials and Methods
Line 102: GIS software to establish coordinates should be stated
Response: Thank you very much for pointing this out. We addressed your comment, and we have added a study areas map (Figure 1).
Line 200-202: Briefly state the method not only cite it
Response: Thank you very much. We followed your suggestion. A description of the method has been added in chemical analysis section 2.6.
Line 386-390: "a diverse community of microbes resilient to toxic metals and contribute to the remediation of wastewater. Soil microbes are located in the close vicinity of plant roots and most of them enter cortical root cells to act as endophytes which facilitate plant growth" This, as previously mentioned, should be further detailed as it is an important and relevant chapter of the presented work.
Response: Dear Reviewer, we have added some more explanation about root exudates in the Discussion (lines 623-666). Other Reviewers already commented that the discussion is long that’s why we tried to wind up the discussion. We tried to satisfy the requirements of all Reviewers and provide a reasonable balance between the length of our paper and additional information.
Thank you very much once again for your essential comments and support while improving the scientific value of this manuscript. We hope that implemented corrections will be gratifying for you in making a favourable decision about it.
References
- Peuke AD, Rennenberg H. Phytoremediation. EMBO Rep. 2005;6(6):497-501. doi:10.1038/sj.embor.7400445
- FENDRIHAN S, POP CE. Biotechnological potential of
plant associated microorganisms. Rom Biotechnol Lett. 2021; 26(3): 2700-2706. DOI: 10.25083/rbl/26.3/2700-2706
Round 2
Reviewer 3 Report
The manuscript has been improved
I would still ask the authors to be very careful regarding English language and text style because there are still mistakes regarding English language and/or carelessness
please see some examples
replace
The Jokipohja plant 107
contains a mechanical handling of wastewaters, a chemical removal of Phosphorus (P) 108
with ferrous sulphate (FeSO4), polymers and lime (CaO), and a biologically activated 109
sludge handling. The local stainless steel metal industry and the Jyri landfill area of the 110
Outokumpu transfers their wastewaters to the plant for purification
with
The Jokipohja plant 107
is characterized by initial mechanical treatment of wastewaters, chemical removal of P 108
with ferrous sulphate (FeSO4), polymers and lime (CaO), and secondary treatment of activated 109
sludge. The local stainless steel metal industry and the Jyri landfill area of the 110
Outokumpu also transfer their wastewaters to the plant for purification
etc...
2) for all the apparatuses used please give manufacturer, city and country of origin. for example
flow injection analysis (FIA, IO Analytical Flow Solution FS 3700)
so what is the apparatus used? what does FIA stands for? if FS 3700 is the model of the apparatus give also manufacturer, city, country of origin
an inductively coupled plasma-mass spectrometry 213
(ICP-MS, Agilent 7500 Series, Waltham, MA, USA), Perkin Elmer Optima 5300 DV spec- 214
trometer (ICP-OES)
so what is the apparatus used? ICP-MS or ICP-OES? give also manufacturer, city, country of origin
burning samples in a 224
muffle furnace at 450 °C within 8 hours
give all details of furnace , model, manufacturer, city, country of origin
CEM-Mars 5 microwave oven give manufacturer, city, country of origin
ICP-OES, IRIS Intrepid II 234
XSP) give manufacturer, city, country of origin
etc...
Author Response
Dear Reviewer,
Thank you again for helping in manuscript improvement.
The Jokipohja plant 107
contains a mechanical handling of wastewaters, a chemical removal of Phosphorus (P) 108
with ferrous sulphate (FeSO4), polymers and lime (CaO), and a biologically activated 109
sludge handling. The local stainless steel metal industry and the Jyri landfill area of the 110
Outokumpu transfers their wastewaters to the plant for purification
with
The Jokipohja plant 107
is characterized by initial mechanical treatment of wastewaters, chemical removal of P 108
with ferrous sulphate (FeSO4), polymers and lime (CaO), and secondary treatment of activated 109
sludge. The local stainless steel metal industry and the Jyri landfill area of the 110
Outokumpu also transfers their wastewaters to the plant for purification.
Response: It has been revised.
2) for all the apparatuses used please give manufacturer, city and country of origin. for example
flow injection analysis (FIA, IO Analytical Flow Solution FS 3700)
so what is the apparatus used? what does FIA stands for? if FS 3700 is the model of the apparatus give also manufacturer, city, country of origin.
Response: It has been revised. And FIA stands for Flow Injection Analysis.
an inductively coupled plasma-mass spectrometry 213
(ICP-MS, Agilent 7500 Series, Waltham, MA, USA), Perkin Elmer Optima 5300 DV spec- 214 trometer (ICP-OES)
so what is the apparatus used? ICP-MS or ICP-OES? give also manufacturer, city, country of origin
burning samples in a 224
muffle furnace at 450 °C within 8 hours
give all details of furnace , model, manufacturer, city, country of origin
CEM-Mars 5 microwave oven give manufacturer, city, country of origin
ICP-OES, IRIS Intrepid II 234
XSP) give manufacturer, city, country of origin
Response: We have revised all the instrumental information in the materials and methods section 2.6.
Reviewer 4 Report
The authors have made the required modification and I now find the article suitable for publication
Author Response
Dear Reviewer,
Thank you again for helping in manuscript improvement.